# Proton transfer regulated photocured robust room-temperature phosphorescence from naphthalimide

Aicheng Wang[1], Haoxuan Wei[2], Kunquan Lin[1], Xing Huang[3], Mingxing Chen[4], Wentao Bian[1], Junxiao Wang[2], Yuzhou Qiao[1], Bing Fang ®[1]✉, Yuxia Zhao[3], Jianxiang Yu[1], Meizhen Yin ®[2]✉ & Yuhua Dai ®[1]✉

Photocured room-temperature phosphorescence (RTP) materials have considerable potential applications but are rarely reported. Here, we reported photocured RTP materials from naphthalimide, which simultaneously acts as RTP chromophore and photo-initiator. Specifically, naphthalimide generates radicals to polymerize acrylic acid and acrylamide upon UV irradiation. The resulting naphthalimide is tightly restricted in in-situ formed crosslinked matrix to achieve robust RTP ($\tau_p$ = 389.58 ms, $\varphi_p$ = 17.83%, water and organic solvents resistance). Significantly, carboxyl can bind onto lone-pair electrons of tertiary amine in naphthalimide through proton transfer hydrogen-bonds (PTHBs), inhibiting nonradiative decay of $S_1$ induced by photoinduced electron transfer (PET); increasing spin-orbit coupling (SOC) to promote intersystem crossing (ISC); cooperating with intermolecular hydrogen-bonds afford rigid microenvironment to stabilize triplet excitons. Moreover, afterglow colors are continuously tuned after loading different mass RhB via energy transfer. The as prepared materials are used as RTP inks for fabricating 3D printing and photopatterning for anti-counterfeiting and information encryption applications.

Room-temperature phosphorescence (RTP) materials, due to large Stokes shift, long lifetime, and high signal-to-noise ratio, have attracted great attention in the applications of biologic imaging, visual dictation, anti-counterfeiting, and information encryption[1–9]. Until now, organic RTP materials with simple fabrication protocol, ease of processing, lower cost, and flexible mechanical properties have been developed to replace inorganic RTP materials, which often contain high toxicity, high cost, poor-biocompatibility, and complex fabrication processing[10–15]. Generally, the generation of RTP requires the fulfillment of two prerequisites in organic systems: (1) ISC from the lowest excited singlet state ($S_1$) to a triplet state and (2) radiative transitions

from the lowest excited triplet state ($T_1$) to the ground state ($S_0$). Unfortunately, inefficient ISC and triplet excitons quenching directly result in low efficient RTP, largely restricting the development of RTP materials in the fields of optoelectronics[16]. Strategies to develop high-performance organic RTP materials include heavy-atom or lone-pair electron heteroatom effect to strengthen SOC interactions for efficient ISC[17–20]; crystalline engineering, host-guest inclusion, matrix rigidification, and polymerization afford rigid surrounding environment to stabilize triplet excitons for phosphorescence emission[21–27]. Among these, polymerization is a simple fabrication protocol and ease of processing to afford rigid environment to stabilize triplet excitons[28–32].

[1]Beijing Key Laboratory of Special Elastomer Composite Materials, College of New Materials and Chemical Engineering, Beijing Institute of Petrochemical Technology, Beijing, China. [2]State Key Laboratory of Chemical Resource Engineering, Beijing University of Chemical Technology, Beijing, China. [3]Key Laboratory of Photochemical Conversion and Optoelectronic Materials, Technical Institute of Physics and Chemistry, Chinese Academy of Sciences, Beijing, China. [4]Analytical Instrumentation Center of Peking, Peking University, Beijing, China. ✉e-mail: fangbing@bipt.edu.cn; yinmz@mail.buct.edu.cn; daiyuhua@bipt.edu.cn

Significantly, polymerization enables diverse regulations of RTP characteristics (intensity, lifetime, and quantum yield) through gradually rigidity of surrounding environments, which is beneficial for understanding the intrinsic mechanism of RTP[33–35]. Specifically, photocured materials that transfer from a liquid state to a solid state when irradiated with UV light are acknowledged as a familiar type of polymerization system[36,37]. However, photocured RTP system requires phosphorescent chromophore and photo-initiator simultaneously, increasing the stability and complexity of the whole system. Moreover, photo-initiator happens photochemical reaction and may generate adversely factors to quench RTP emission from phosphorescent chromophore. Several groups have reported photocured RTP materials. For example, Chen et al. reported photocured RTP materials using a combination of lignosulfonate, acrylamide, and ionic liquid with an emission lifetime of 110 ms and phosphorescence quantum yield of 11.04%[38]. Chen et al. also reported a solvent free system consisting 2-hydroxyethyl acrylate, urethane dimethacrylate for converting lignin into RTP materials, with an emission lifetime of 202.9 ms and humidity/water-resistant performance[39]. Ma et al. reported cycloaliphatic epoxy resin materials by utilizing three phosphors as hydrogen stripping initiators and the obtained polymer systems perform optimally in terms of photocuring property, mechanical property, and solvent resistance with controllable RTP emission[40]. Comfortingly, some photocured RTP systems have been developed, albeit with limited RTP performance. Therefore, it is of an urgent demand to develop high-performance photocured RTP systems integrating with phosphorescent chromophores and photo-initiators.

Naphthalimide is a hydrogen stripping photo-initiator and often requires the cooperation of tertiary amines as proton donor to increase efficiency and stability[41], however, tertiary amines severely suppress the optical properties of naphthalimide according to nonradiative decay of $S_1$ state, which are induced by photoinduced electron transfer (PET) effect[42]. Nonradiative decay of $S_1$ state severely restricts ISC process, and largely limited generates triplet excitons for phosphorescence emission. Therefore, in the naphthalimide system, a productive strategy is to introduce a photophysical process to inhibit PET, thereby enabling more $S_1$ excitons to undergo ISC to generate triplet excitons for phosphorescence emission, and meanwhile retaining photo-initiating ability. Herein, we introduced tertiary amines into naphthalimide (NDIAM) to simultaneously act as phosphorescent chromophore and photo-initiator. Specifically, NDIAM generates radicals to polymerize acrylic acid (AA), and acrylamide (AM) upon UV irradiation. As a result, NDIAM is immobilized in the rigid microenvironment to generate robust RTP with water and organic solvents resistance, long lifetime of 389.58 ms and high phosphorescence quantum yield of 17.83% (Fig. 1). Significantly, carboxyl group can bind onto the lone-pair electrons of tertiary amine in NDIAM through proton transfer hydrogen-bonds (PTHBs)[43], which play as following roles: (1) inhibit nonradiative decay of $S_1$ state induced by PET; (2) increase SOC values to promote ISC; (3) cooperate with intermolecular hydrogen-bonds affording rigid microenvironment to stabilize triplet state excitons. Control experiments, such as photo-polymerizing different ability of proton-donating monomer (styrene, methacrylate, methyl methacrylate, hydroxyethyl acrylate, and AA; photo-copolymerizing these monomers with AA; dispersing NDIAM into PVA with or without acid, they all show yellow afterglows when NDIAM meets with acid, which illustrates PTHBs playing a crucial role in regulating RTP characteristic. This work provides an efficient method to achieve high-performance RTP in photocured system and has a great potential in application of 3D printing and photopatterning.

## Results

### PTHBs regulated photocured robust RTP

The naphthalimide derivative bearing tertiary amine group was successfully synthesized using N, N-dimethylethylenediamine and 1,8-naphthalic anhydride, and named as NDIAM (Supplementary Fig. S1)[44]. The structure and purity were confirmed by ¹H NMR, ¹³C NMR, MS, and HPLC spectroscopic analyses (Supplementary Figs. S2–5 and Supplementary Table 1). NDIAM exhibits absorption characteristics in the UV–vis spectra range in 200–400 nm, confirming its capacity to absorb 365 nm UV light and thereby enabling its utilization in subsequent steps (Supplementary Fig. S6). The ESR spectrum shows silently signal before 365 nm UV light irradiation. After 365 nm UV light irradiation for 20 s and 35 s (Power intensity: 300 mW/cm²), obvious photochemical free radicals appear (Fig. 2b)[45]. These generated photochemical free radicals can be used to initiate free radical polymerization of acrylic ester compounds. Therefore, the mixed solution of NDIAM, AA, and AM (NDIAM-AA-AM, 1 mg NDIAM, 400 μL AA, and 500 mg AM) are continuously exposed to UV light for photocuring (P-AA-AM). The photocuring process is monitored by FT-IR, and the double bond conversion rate reaches ~91.80% after 365 nm UV light irradiation for 40 s (Fig. 2c, Power intensity: 300 mW/cm²)[40]. Increasing the content of NDIAM (from 1 mg to 3 mg, AA and AM remain unchanged), UV photocured time for the double bond conversion rate reaching maximum is shorten to 25 s (Supplementary Fig. S7). To demonstrate the generation of free radicals during the photocuring process, 5,5-dimethyl-1-pyrroline N-oxide (DMPO) was incorporated into the photocuring formulation as a free radical scavenger, with a control experiment conducted in its absence. As anticipated, in the presence of DMPO, no polymerization was observed after 3 min of UV irradiation. In contrast, when DMPO is absent, NDIAM-AA-AM fully photocured under UV irradiation for 30 s (Supplementary Fig. S8). To understand the reaction, the precursor solution of NDIAM-AA-AM was analyzed by ¹H NMR before and after UV irradiation. The results show that the signal of the double bond decrease, indicating that the photopolymerization occurs Supplementary Fig. S9). In addition, P-AA-AM demonstrates high transparency across the visible spectrum, high mechanical strength, abundant intermolecular H-bonds interactions, and amorphous molecular packing structure (Supplementary Fig. S10). Glass transition temperature of P-AA-AM is 173.83 °C (Supplementary Fig. S11). Following, the photophysical properties during photocured process are systematically explored. NDIAM-AA-AM exhibits no RTP emission before photocuring (Supplementary Figs. S12 and 13). This is attributed to the high nonradiative decay rate in the liquid environment around NDIAM, thereby causing RTP quenching[46]. However, both RTP intensity and lifetime sharply enhance with the extension of photocuring time (from 0 to 50 s) (Fig. 2d–g), which are caused by a decrease in nonradiative decay with gradually increased rigid microenvironment. Upon photocuring process, P-AA-AM achieves a prominent transformation from the no afterglow to a bright 3 s yellow afterglow (Fig. 2a). As such, lifetime increases to 389.58 ms ($\tau_p$), and phosphorescence quantum yield reaches as high as 17.83% ($\varphi_p$) after photocuring for 50 s (Supplementary Fig. S14). When temperature lowered from 298 to 77 K, phosphorescence intensity varied greatly, which is due to the nonradiative decay is greatly suppressed (Supplementary Fig. S15). The influence of mass of NDIAM on phosphorescent intensity and lifetime was also investigated. 0.01, 0.1, 1, 5, and 10 mg NDIAM were used to photo-polymerize AA and AM. Phosphorescence intensity and lifetime reach a maximum at 1 mg NDIAM (Supplementary Fig. S16). It could be concluded that excess NDIAM resulted in aggregation-caused quenching and energy dissipation[14]. It's worth mentioned that P-AA-AM shows high-performance RTP with simultaneously high $\varphi_p$ and long $\tau_p$, which is a big challenge to achieve in photocured organic RTP field.

Conformingly, P-AA-AM showed robust RTP characteristic with water and organic solvents resistance, which had rarely been observed for similar systems. Specifically, 233.61 ms long lifetime, strong phosphorescence intensity, and 1.5 s yellow afterglow are still observed when it immerses in water for 210 min. After it immerses in water for 240 min, RTP lifetime decreases to 93.70 ms, and yellow afterglow

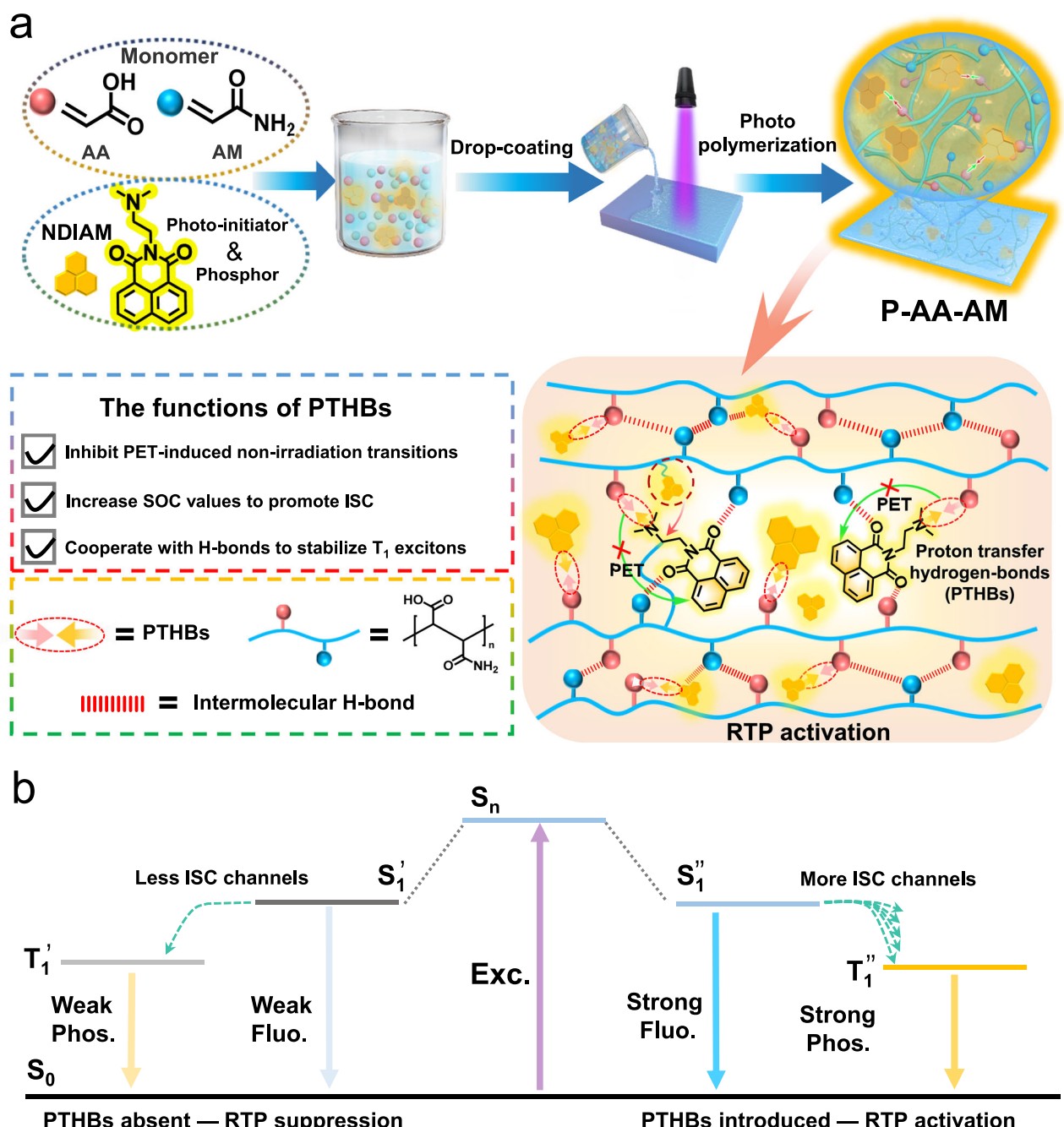

**Fig. 1 | Introduction of phosphor and photocuring ability. a** Scheme illustrating photocured RTP materials from naphthalimide (PET photoinduced electron transfer, RTP room-temperature phosphorescence, PTHBs proton transfer hydrogen-bonds); **b** Mechanism for PTHBs regulated RTP performance (Fluo. fluorescence, Phos. phosphorescence, ISC intersystem crossing, Exc. Excitation).

disappears (Fig. 2h, i and Supplementary Fig. S17). The RTP intensity and lifetime of P-AA-AM can restore to its original state after removing the water (Supplementary Fig. S18). To understand the phenomenon, the contact angle and the surface tension of P-AA-AM were measured (Supplementary Fig. S19). The value of contact angle and the surface tension are 63.2° and 21.6 mN/m, indicating the characteristic of hydrophilicity. This is because the water molecules form H-bonds with the surface carboxyl and amido bond in P-AA-AM. Such interactions induced formation of dense molecular networks, decelerating water molecules entering the network. As a result, the triplet excitons are well protected in P-AA-AM and RTP emission is stable when the sample is immersed into water. The stability of P-AA-AM is further investigated

in organic solvents, RTP lifetime and intensity is not compromised when it is immersed into different organic solvents, such as, n-Hexane, dimethyl formamide (DMF), dichloromethane, methyl tert-butyl ether, tetrahydrofuran, and ethyl alcohol for 45 days (Fig. 2j and Supplementary Fig. S20). As a result, robust RTP with water and organic solvents resistance is successfully achieved.

To get more insight into mechanism of photocured robust RTP of P-AA-AM, series of control experiments were designed. Firstly, NDIAM is dissolved into different proton donating ability of ethylene monomer, such as styrene (St), methacrylate (MA), methyl methacrylate (MMA), hydroxyethyl acrylate (HEA), and AA (named as NDIAM-St, NDIAM-MA, NDIAM-MMA, NDIAM-HEA, and NDIAM-AA, respectively).

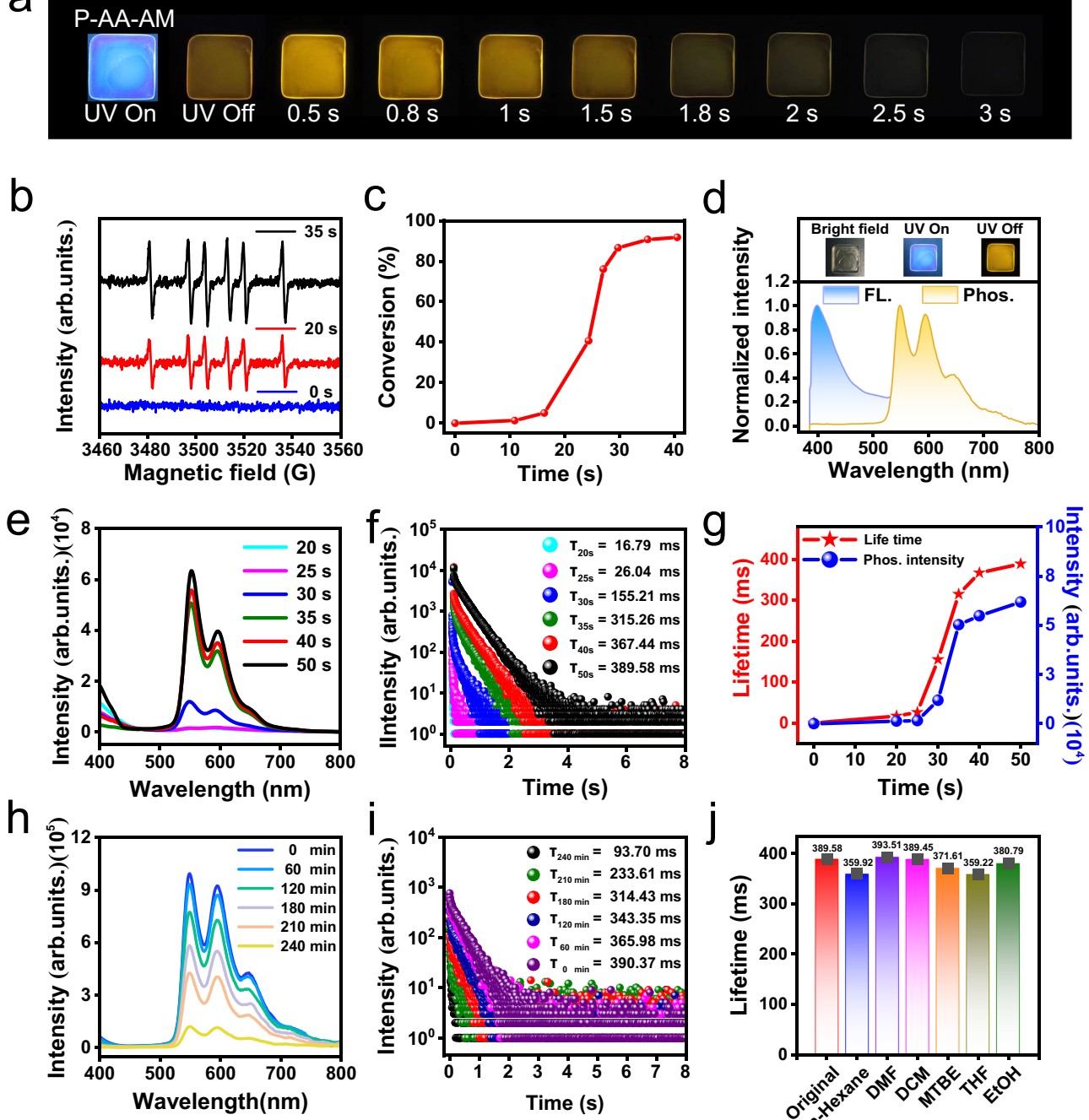

**Fig. 2 | Photophysical properties of P-AA-AM. a** Afterglow photographs of P-AA-AM film; **b** The ESR spectra of NDIAM in AA under 365 nm UV irradiation: 0 s (blue line), 20 s (red line) and 35 s (black line); **c** The double bond conversion of NDIAM-AA-AM under 365 nm UV irradiation different time; **d** The fluorescence (FL) and phosphorescence (Phos.) spectra of P-AA-AM, Inset: images of P-AA-AM under visible light (left), UV light (middle), and UV off (right); **e** RTP spectra, **f** RTP lifetime ($\lambda_{collected}$ = 550 nm) and **g** summary of RTP property of P-AA-AM in the photocuring process; **h** RTP spectra, RTP lifetime of P-AA-AM after (**i**) various water-soaking durations and **j** different organic solvents (single measurement). $\lambda_{ex}$ = 365 nm, delay time = 1 ms.

St, MA, MMA, and HEA did not have active hydrogen atom to bind onto the lone-pair electrons of tertiary amine. All these solutions form solid film on 2 × 2 cm glass (named as P-St, P-MA, P-MMA, P-HEA, and P-AA) under 365 nm UV light irradiation. 1.5 s yellow afterglow is only observed in P-AA after removal 365 nm UV light (Fig. 3a and Supplementary Fig. S21); P-St, P-MA, P-MMA, and P-HEA show no afterglow appearance (Fig. 3a). The double bond conversion rates reach ∼ 22.22%, 66.03%, 84.44%, 69.07%, and 61.78% in P-St, P-MA, P-MMA, P-HEA, and P-AA, respectively, after 365 nm UV light irradiation for 300 s

(Supplementary Fig. S22). The prompt and delayed spectra of P-AA shows much higher fluorescence and phosphorescence intensity than that of P-St, P-MA, P-MMA, and P-HEA (Fig. 3b and Supplementary Fig. S23a). The lifetime decay curve of P-AA is used to certificate RTP and shows long lifetime of 156.79 ms. The lifetime decay curve of P-St, P-MA, P-MMA, and P-HEA are 23.26, 21.45, 40.25, and 48.21 ms, respectively, which are much shorter than that of P-AA (Fig. 3c). The phosphorescence quantum yield of P-AA is 12.46%, is also much higher than that of P-St, P-MA, P-MMA, and P-HEA (0.87%, 0.51%, 0.56%, and

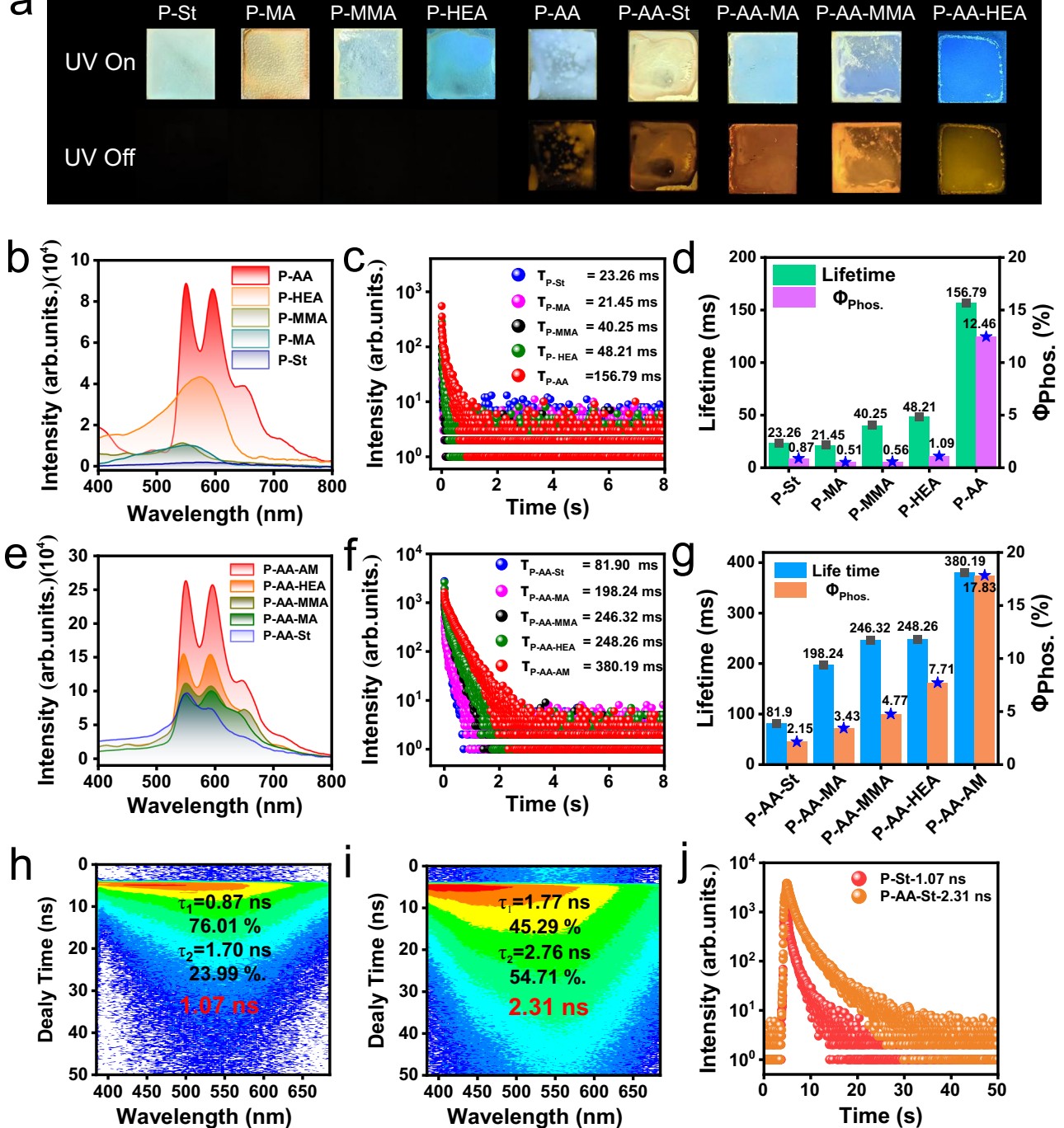

**Fig. 3 | Photophysical properties of photocured materials. a** The photographs of photocured different monomer and copolymerize these monomers with AA; **b** The delayed spectra and **c** Phosphorescence lifetime of photocured material from different monomer ($\lambda_{ex}$ = 365 nm, delay time = 1 ms, $\lambda_{collected}$ = 550 nm); **d** Phosphorescence (Phos.) lifetime and quantum yield of photocured material (single measurement); **e** Delayed spectra, **f** phosphorescence lifetime of copolymerizing monomer with AA ($\lambda_{ex}$ = 365 nm, $\lambda_{collected}$ = 550 nm, delay time = 1 ms); **g** Summary phos. lifetime and phos. quantum yield of photocured materials (single measurement); Time-resolved fluorescence spectra of **h** P-St and **i** P-AA-St; **j** Average fluorescence lifetime of P-St and P-AA-St.

1.09%, respectively) (Fig. 3d, Supplementary Fig. S24, and Table 2). Together with phosphorescence intensity, phosphorescence lifetime, and phosphorescence quantum yield, P-AA shows the best RTP performance among these polymers. These results indicate that PET quenches RTP performance in P-St, P-MA, P-MMA, and P-HEA, however, carboxyl of P-AA form PTHBs with tertiary amine of NDIAM to suppress nonradiative decay of $S_1$ state induced by PET. Meanwhile, PTHBs and intermolecular H-bonds immobilize NDIAM in P-AA and

stabilize triplet excitons, thereby resulting in the appearance of the best RTP performance among these homopolymers.

Next, we use AA to photo-copolymerize with St, MA, MMA, and HEA (named as P-AA-St, P-AA-MA, P-AA-MMA, and P-AA-HEA, respectively) to elucidate PTHBs playing a crucial role in regulating RTP performance. After removal 365 nm UV light irradiation, bright yellow afterglows are all obtained in P-AA-St, P-AA-MA, P-AA-MMA, and P-AA-HEA film (Fig. 3a). The delayed spectra of P-AA-St, P-AA-MA,

P-AA-MMA, and P-AA-HEA show phosphorescence peaks at 550 and 600 nm (delay time = 1 ms) (Fig. 3e) and exhibit sharply enhanced fluorescence and phosphorescence intensity when compared with that of P-St, P-MA, P-MMA, and P-HEA (Supplementary Figs. S23b, S25–28). P-AA-St, P-AA-MA, P-AA-MMA, and P-AA-HEA also exhibit prolonged RTP lifetime of 81.90, 198.24, 246.32, and 248.26 ms, respectively (Fig. 3f, g), (3.52 times for P-AA-St, 9.24 times for P-AA-MA, 6.12 times for P-AA-MMA and 5.15 times for P-AA-HEA) (Supplementary Figs. S25–28) and enhanced phosphorescence quantum yield of 2.15%, 3.43%, 4.77%, and 7.71%, respectively (4.21 times for P-AA-St, 9.82 times for P-AA-MA, 9.48 times for P-AA-MMA and 6.72 times for P-AA-HEA, respectively) (Supplementary Figs. S25–28). The double bond conversion rate reaches ~ 77.08%, 90.96%, 85.44%, and 88.73% in P-AA-St, P-AA-MA, P-AA-MMA, and P-AA-HEA, respectively, after 365 nm UV light irradiation (Supplementary Fig. S29, Supporting Information). P-AA-St and P-St were selected to test time-resolved fluorescence, the excited-state lifetime in P-AA-St was longer than that of P-St (Fig. 3h–j and Supplementary Fig. S30), indicating that PTHBs between acid-amine in P-AA-St effectively suppressed nonradiative $S_1$ decay, thereby resulting in the appearance of sharply enhanced phosphorescence lifetime and phosphorescence quantum yield. In addition, the phosphorescence intensity, phosphorescence lifetime, and phosphorescence quantum yield of P-AA-St, P-AA-MA, P-AA-MMA, and P-AA-HEA are lower than that of P-AA-AM. This is because carboxyl and amido bond in P-AA-AM form stronger intermolecular H-bonds, providing a more rigid microenvironment that effectively immobilizing NDIAM motion to stabilize triplet excitons for phosphorescence emission.

Polyvinyl alcohol (PVA) serves as an ideal matrix owing to its abundant intermolecular H-bonds for stabilizing triplet excitons, and has achieved rapid development in the field of RTP[47–50]. We dispersed 1 mg NDIAM into without or with carboxyl compounds (such as DMF, butyric acid (BuA), propionic acid (PA), AA, acetic acid (AcA), and benzoic acid (BA)), then quickly poured into 100 mg/mL PVA water solutions and dripped coating for film formation (named as NDIAM-DMF/PVA, NDIAM-BuA/PVA, NDIAM-PA/PVA, NDIAM-AA/PVA, NDIAM-AcA/PVA, and NDIAM-BA/PVA). NDIAM-DMF/PVA shows weak fluorescence and no afterglow, however, added carboxyl compounds samples all exhibit strong blue fluorescence and yellow afterglow, lasting for more than 1 s (Supplementary Fig. S31). The phosphorescence intensity of added carboxyl compounds samples is significantly increased when compared with that of NDIAM-DMF/PVA (Supplementary Fig. S32a). The lifetime of NDIAM-DMF/PVA can't be fitted due to low phosphorescence intensity, however, the lifetime decay curve of NDIAM-BuA/PVA, NDIAM-PA/PVA, NDIAM-AA/PVA, NDIAM-AcA/PVA, and NDIAM-BA/PVA show long lifetime of 119.05, 179.20, 200.42, 222.10, and 274.96 ms, respectively (Supplementary Fig. S32b). To further investigate the role of acids in regulating RTP performance, phosphorescence properties of NDIAM-DMF/PVA and NDIAM-AA/PVA under vacuum and low-temperature conditions are conducted. Under vacuum condition, phosphorescence intensity and lifetime of NDIAM-DMF/PVA are too low to detect; strong phosphorescence intensity and 210.18 ms lifetime are obtained in NDIAM-AA/PVA (Supplementary Fig. S33). It was not oxygen that quenched the phosphorescence of NDIAM-DMF/PVA. Under low-temperature condition, NDIAM-AA/PVA shows higher phosphorescence intensity and longer lifetime than that of NDIAM-DMF/PVA (Supplementary Figs. S34 and 35). These results indicate that PTHBs between acid-amine promoting triplet formation and stabilizing triplet state for phosphorescence emission. Protonated NDIAM was prepared by adding HCl into NDIAM solution, and then evaporating solvents (NDIAM-H). We doped NDIAM-H into γ-cyclodextrin (γ-CD) and polyacrylic acid (PAA). Stronger phosphorescence intensity and longer lifetime of NDIAM-H/PAA than that of NDIAM-H/γ-CD are obtained, indicating that physical effect of matrix rigidification also

play an important role in regulating RTP performance (Supplementary Fig. S36). Consequently, high-performance RTP can be realized by the combinations of PTHBs and intermolecular hydrogen bonds, which facilitate triplet formation and stabilize triplet state for phosphorescence emission.

## Mechanism for PTHBs regulated RTP performance

To investigate the mechanism of PTHBs regulated RTP performance, the absorption, FTIR and $^1$H NMR of NDIAM before and after adding AA solution were carried out. The absorption peak at ~ 235 nm exhibits gradually reduced trend upon adding AA into NDIAM MeOH solution (Supplementary Fig. S37). Upon adding NDIAM and triethylamine into AA solution, the vibration peaks of the hydroxyl at 2992 cm$^{-1}$ and the carbonyl at 1694 cm$^{-1}$ in the carboxyl weaken; while the vibration peaks of the asymmetric and symmetric the carboxylate at 1541 cm$^{-1}$ and 1355 cm$^{-1}$ appear (Supplementary Fig. S38). The hydrogen atoms in methyl group of NDIAM, exhibit significant chemical shift upon adding AA (Supplementary Fig. S39). These changes upon NDIAM meeting with AA, show the formation of PTHBs between the tertiary amine and carboxyl groups. We also carried out a series of theoretical calculations between various homopolymers and NDIAM. All molecular conformations were optimized and vibrational frequency analyses were performed via the Gaussian 09 program at the B3LYP/6-31 G(d,p) level of theory. The electrostatic surface potential (ESP) analysis indicates that tertiary amine and carbonyl of NDIAM show strong negative electrostatic potential (−31.98 and −34.91 kcal/mol, respectively), carboxyl group of PAA show positive ESP values (62.19 kcal/mol), which indicating strong intermolecular interactions with NDIAM (Fig. 4a and Supplementary Fig. S40)[51]. According to the independent gradient model based on Hirshfeld partition (IGMH) analysis (Fig. 4b and Supplementary Figs. S41–43), the intermolecular interactions in NDIAM/homopolymers are dominated by attractive interactions and van der Waals interaction[52]. For NDIAM/PAA, the carboxyl groups of PAA can form strong H-bonds by proton transfer with tertiary amine in NDIAM, which can influence energy level of tertiary amine; there also exist H-bonds between the carboxyl groups of PAA and carbonyl of NDIAM. To gain a clearer understanding of proton transfer regulating energy alignment and orbital characteristics, we employ the Gaussian 09 software package and adopt the B3LYP/6-311G* basis set to perform density-functional theory (DFT) and time-dependent DFT (TD-DFT) calculations. The HOMO and LUMO of trimethylamine (TMA) overlap with that of naphthalimide without tertiary amine (NA), indicating PET would happen between TMA and NA moiety. NDIAM exhibits weak fluorescence and phosphorescence, due to $S_1$ excitons adopting nonradiative decay induced by PET (Supplementary Fig. S44). Following, energies of frontier orbitals of TMA interact with various monomers (TMA-St, TMA-MA, TMA-MMA, TMA-HEA, and TMA-AA) and NA are calculated. The HOMO and LUMO energy of TMA-St, TMA-MA, TMA-MMA, and TMA-HEA are still overlapped with than that of NA, indicating these monomers can't suppress nonradiative decay of $S_1$ state induced by PET. However, AA can effectively bind onto the lone-pair electrons of TMA through PTHBs, leading to significantly affect the energy level distribution of TMA. The HOMO and LUMO energy of TMA-AA include that of NA, exhibiting nonradiative decay of $S_1$ state induced by PET are effectively suppress, leading to more $S_1$ excitons to undergo ISC to generate triplet excitons (Fig. 4c). To demonstrate their differences in ISC, energy levels and SOC constants are calculated by ORCA4.2.0 (Fig. 4d, Supplementary Fig. S45, and Supplementary Table 5). Within the ± 0.3 eV range of $S_1$ energy level, NDIAM/PAA exhibits the probability of ISC to $T_5$–$T_9$ states with large SOC constants of 0.20, 0.30, 0.78, 1.59, 0.81, and 0.21 cm$^{-1}$, respectively. While the other homopolymer chains exhibit less ISC channels ($T_2$ for NDIAM, NDIAM/PSt, and NDIAM/PMA; $T_2$ and $T_3$ for NDIAM/PHEA), and small SOC values

(0.18 cm$^{-1}$ for NDIAM; 0.16 cm$^{-1}$ for NDIAM/PSt, 0.48 cm$^{-1}$ for NDIAM/PMA; 0.15 and 0.16 cm$^{-1}$ for NDIAM/PHEA). The increasing ISC channels and larger SOC values in NDIAM/PAA demonstrate PTHBs largely promote ISC for enhancing RTP performance. The natural transition orbitals (NTOs) of NDIAM in PSt, PMA, PHEA, and PAA are also calculated (Fig. 4e). For $S_1$ state as example, the hole and the electron distribute on PAA in NDIAM/PAA; while the hole and the electron distribute on NDIAM in NDIAM, NDIAM/PSt, NDIAM/PMA, and NDIAM/PHEA. The complete transfer of the hole and the electron verify a charge transfer of $S_1$ state in NDIAM/PAA, which promotes ISC from $S_1$ to $T_m$ and larger ISC constant ($k_{ISC}$) (Supplementary Fig. S46)[53]. The $k_{ISC}$ of NDIAM/PAA is as large as $6.29 \times 10^7$, which is higher than that of NDIAM/PSt, NDIAM/PMA, and NDIAM/PHEA (Supplementary Fig. S47 and Supplementary Tables S2 and 3). Meanwhile, the hole and the electron of $T_1$ in NDIAM/PAA are also distributed into PAA, corresponds to a charge transfer $T_1$ state, leading to a small phosphorescent nonradiative decay constant ($k_{p,nr}$) (Supplementary Fig. S48). The $k_{p,nr}$ of P-AA is 5.07 s$^{-1}$ and is much lower than that of P-St, P-MA, P-MMA, and P-HEA. $\tau_p$ of P-AA could be extended by the equation of $\tau_p = 1/(k_{p,r} + k_{p,nr})$[54]. As a result, the introduction of PTHBs in NDIAM photocured materials can inhibit nonradiative decay of $S_1$ state induced by PET and increase SOC values to promote ISC, leading to long lifetime RTP.

### Tunable afterglow through triplet-to-singlet förster resonance energy transfer

The wide phosphorescence band of P-AA-AM provides a significant advantage when it is used as an energy donor to prepare afterglow materials through the triplet-to-singlet FRET (TS-FRET) process (Fig. 5a). RhB is selected as the ideal acceptor due to the highly overlapped UV absorption spectrum with the phosphorescence spectrum of P-AA-AM (Fig. 5b)[55]. A FRET system with RhB as an energy acceptor and P-AA-AM as an energy donor is constructed (P-AA-AM/RhB, weight ratio of 1:1). As anticipated, the RTP characteristics of P-AA-AM/RhB are significantly different from those of P-AA-AM alone. Specifically, P-AA-AM/RhB exhibits fluorescence emission near ~614 nm, which is attributed to RhB. Additionally, P-AA-AM/RhB displays a delayed RTP emission at around ~610 nm, with a lifetime of approximately 241.58 ms (Fig. 5c). Interestingly, strong delayed emission is observed under indirect excitation at 365 nm, but no obvious delayed emission of RhB was observed under direct excitation at 550 nm (Fig. 5d). These results indicate that P-AA-AM/RhB undergoes an efficient TS-FRET process from P-AA-AM to RhB, and the afterglow of P-AA-AM/RhB in the long wavelength region can be achieved by the delayed fluorescence of RhB. To further understand the energy transfer process, we measure phosphorescence properties of P-AA-AM/RhB after adding different concentrations of RhB in the system, including the changes of fluorescence spectra, phosphorescence spectra, and phosphorescence lifetime (Fig. 5e and Supplementary Figs. S49 and 50). These results show that with change of RhB weight ratio, the delayed spectra intensity at 550 nm of P-AA-AM/RhB system gradually weakens and finally disappears completely upon increasing weight ratio 1:10 (P-AA-AM/RhB). At the same time, the position of delayed spectra is gradually red-shifted from 550 nm to 614 nm. The reason is that RhB molecules are prone to aggregate with lower energy levels at higher concentrations, resulting in red-shifted delayed emission peak. The coordinates of different weight ratio P-AA-AM/RhB = 1/0, 1/0.1, 1/0.5, 1/1, 1/3, 1/5, and 1/10 are (0.44, 0.47), (0.52, 0. 43), (0.56, 0.41), (0.58, 0.38), (0.63, 0.36), (0.64, 0.35), and (0.65, 0.34), respectively, and exhibit gradually red-shifted emission from yellow to red region (Fig. 5f). The time-resolved photoluminescence analysis of P-AA-AM/RhB delayed emission peak at 550 nm ($\lambda_{ex} = 465$ nm) revealed a significant reduction in the average lifetime, from 373.50 ms to 108.40 ms (Fig. 5g). The decreased lifetime provides strong evidence for the non-radiative FRET from the triplet state of P-AA-AM donor to the singlet state of the RhB acceptor in the P-AA-AM/RhB system through the TS-FRET mechanism. The TS-FRET efficiency ($\varphi_{FRET}$) was calculated as 14.8%, 19.0%, 35.3.%, 59.1.%, 68.0%, and 71.0% based on the equation $\varphi_{FRET} = 1 - \tau/\tau_0$, where $\tau$ and $\tau_0$ represented phosphorescence lifetimes of energy donor before and after energy transfer (Fig. 5h and Supplementary Table 4). Therefore, a large range of color adjustment can be conveniently obtained with afterglow, which is conducive to expand their practical applications in many fields.

### 3D printing and photopatterning applications

The precursor solution of NDIAM-AA-AM has good fluidity and can be adaptively changed according to the shape of the curing mold. We pour the precursor solution into the mold of different shapes, and in-situ exposure under UV irradiation, we can easily prepare a variety of desired shapes. Specifically, we first inject the precursor solution into the silica gel mold and obtain a P-AA-AM thin layer with yellow RTP after UV irradiation. The P-AA-AM thin layer will be superimposed and solidified layer by layer through photocuring technology, and finally a 3D bulk material with a pentagonal flower morphology will be constructed. The material exhibits blue fluorescence under 365 nm UV excitation, and exhibits bright yellow afterglow after the UV light turned off (Fig. 6a and Supplementary Fig. S51). Similarly, we use the red afterglow of P-AA-AM/RhB system to make a 3D bulk material with red RTP emission (Fig. 6b). We can also prepare complex 3D bulk materials with both yellow and red RTP emission by layer-by-layer photocuring approach (Fig. 6c). P-AA-AM and P-AA-AM/RhB for UV-curable coating of cotton yarn (Fig. 6d). Specifically, the woven shape of the cotton yarn is immersed in a solution containing the precursor. Then, the treated yarn was subjected to UV irradiation. After P-AA-AM or P-AA-AM/RhB treatment, the coated yarns show yellow or red afterglow emission, respectively. Encouraged by this, these RTP yarns are used to generate cotton thread for anti-counterfeiting purposes using commercial machines. Under 365 nm UV irradiation, the blue and orange red on P-AA-AM cotton and P-AA-AM/RhB cotton are clearly visible. When UV light is turned off, the P-AA-AM/RhB cotton thread remains red afterglow. P-AA-AM cotton thread retains yellow afterglow.

To further demonstrate its application potential, polyurethane acrylate resin is added to the precursor solution of NDIAM-AA-AM or NDIAM-AA-AM/RhB to adjust the viscosity, and ink materials are suitable for photopatterning. The detailed steps of photopatterning process are illustrated in Fig. 6e. The material can be precisely patterned into diverse designs after UV irradiation and water developing, thereby yielding a patterned RTP (Fig. 6f). It is worth noting that the substrate can be flexibly varied, such as glass, PVC film, and PET film. On the PVC film, butterfly pattern can be freely curled and bent without sacrificing its luminescent properties (Supplementary Figs. S52–54). Especially, school emblem and high-precision micro-graphics in the form of QR codes with RTP characteristic is successfully obtained on a glass substrate (Fig. 6g, h).

## Discussion

In conclusion, we have achieved a photocured robust RTP materials with a lifetime of 389.58 ms, phosphorescence quantum yield of 17.83%, and water and organic solvents resistance. NDIAM simultaneously acts as RTP chromophore and photo-initiator in the as-formed material, greatly making the entire process much more convenient. Significantly, PTHBs between carboxyl unit and tertiary amine of NDIAM inhibit nonradiative decay of $S_1$ induced PET; increase SOC constants and ISC channels, generating more $T_1$ excitons. PTHBs and intermolecular hydrogen bonds afford rigid microenvironment to stabilize $T_1$ excitons for RTP emission. Moreover, yellow to red afterglow colors are continuously tuned after

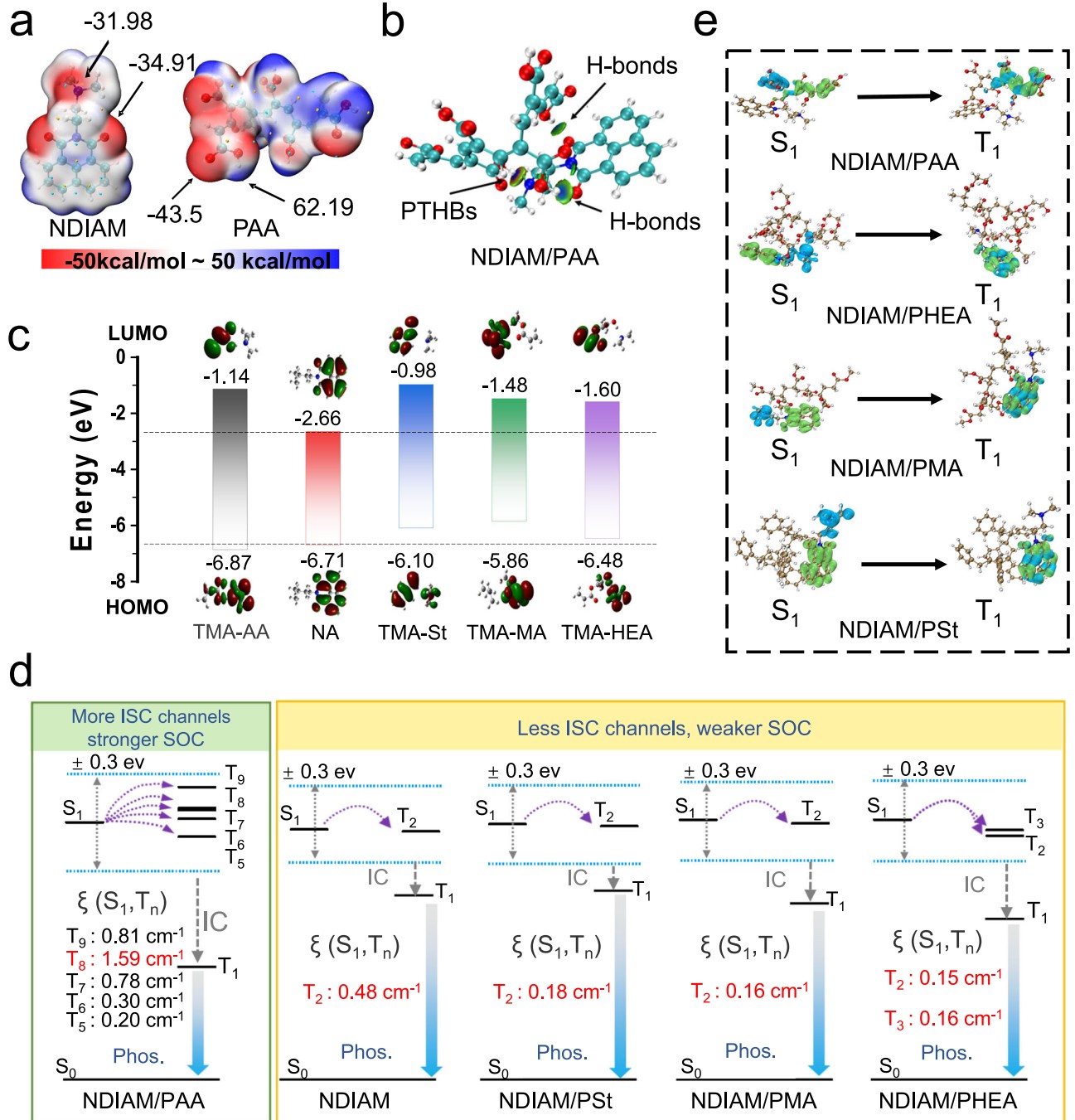

**Fig. 4 | PTHBs regulated RTP mechanism. a** The ESP analysis of NDIAM and P-AA; **b** IGMH scatterplots of NDIAM/PAA; **c** HOMO and LUMO of TMA-AA, NA, TMA-St, TMA-MA, and TMA-HEA; **d** The SOC values of the ISC process in calculation models; **e** NTOs of $S_1$ and $T_1$ in NDIAM/PAA, NDIAM-PHEA, NDIAM/PMA, and NDIAM-PSt (green color: electron, blue color: hole).

loading different mass RhB via TS-FRET strategy. Using these properties, the as prepared materials are used as RTP inks for fabricating 3D printing and photopatterning for anti-counterfeiting and information encryption applications.

## Methods
### Photocuring of P-AA-AM
A mixture containing 400 µL of acrylic acid (5.83 mmol), 500 mg of acrylamide (7.03 mmol), and 1 mg of NDIAM was heated to 60 °C until it formed a homogeneous solution. Subsequently, the solution was subjected to photocuring under a 365 nm UV light with an intensity of 300 mW/cm² for 50 s.

### Photocuring of P-AA-AM/RhB
A solution was prepared by mixing AA (400 µL, 5.83 mmol), AM (500 mg, 7.03 mmol), NDIAM (1 mg, 0.00373 mmol), and RhB (3 mg, 0.00626 mmol). This mixture was heated to 60 °C until it achieved homogeneous solution. Subsequently, the homogeneous solution was subjected to photocuring by exposure to 365 nm UV light with an intensity of 300 mW cm⁻² for 1 min. After 1 min, P-AA-AM/RhB material with red RTP was obtained.

### Preparation of photolithography ink materials
To meet the viscosity requirements for photolithography, the precursor solution of NDIAM-AA-AM was mixed with 150 mg of

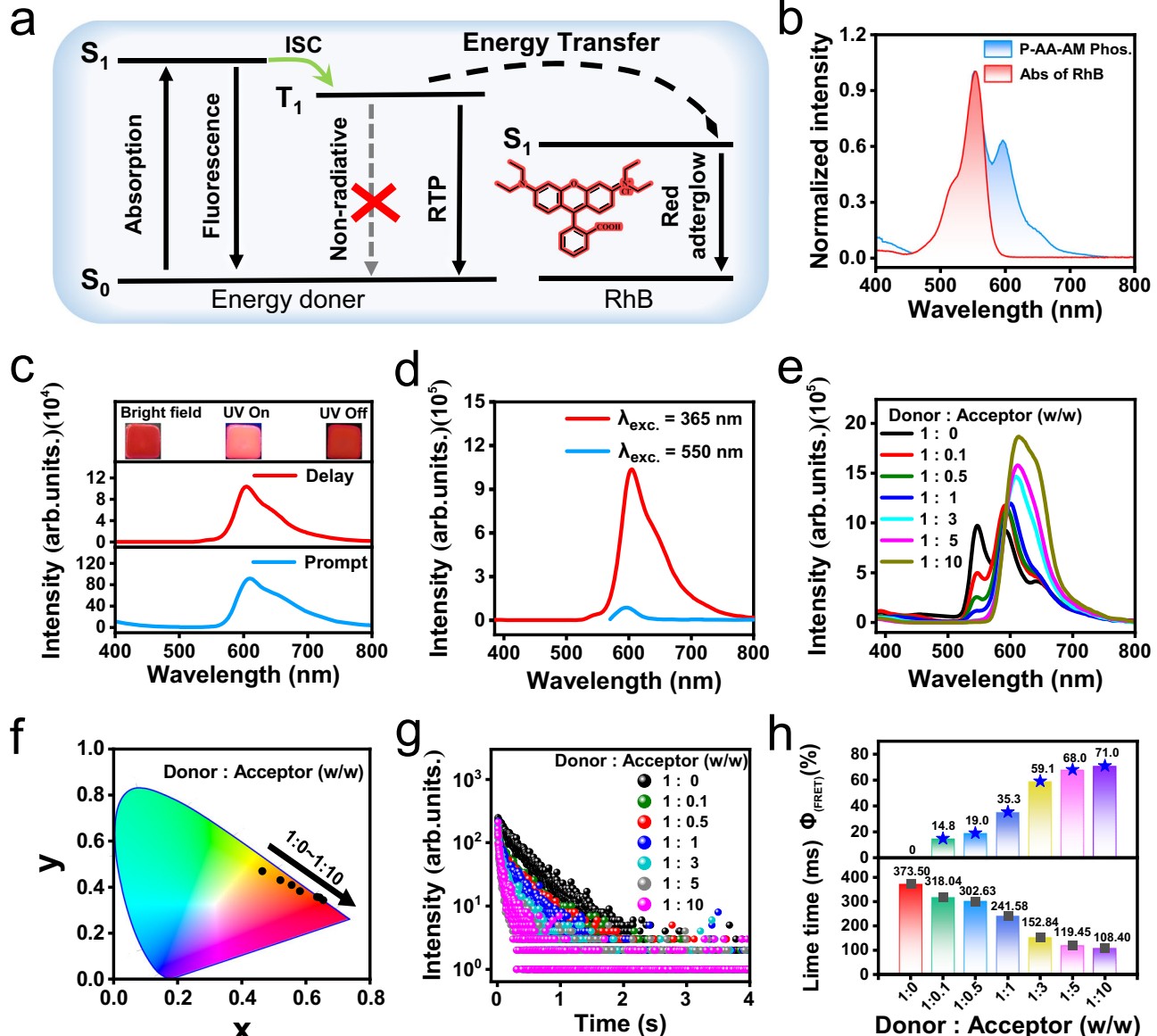

**Fig. 5 | Preparation tunable afterglow of P-AA-AM/RhB. a** Simplified diagram of FRET process; **b** The phosphorescence spectra of P-AA-AM and absorbance of RhB; **c** The prompt and delayed spectra of P-AA-AM/RhB under 365 nm light excitation; **d** The delayed spectra of P-AA-AM/RhB under direct excitation ($\lambda_{ex}$ = 550 nm) and indirect excitation ($\lambda_{ex}$ = 365 nm), Inset: the images of P-AA-AM/RhB under sunlight, UV and after turning off UV; **e** RTP emission, **f** CIE coordinates of P-AA-AM/RhB and **g** RTP lifetime with different mass RhB loadings; **h** Energy-transfer efficiency and phosphorescence lifetime versus stepwise-increased RhB loadings (single measurement).

polyurethane acrylate resin. The mixture was then uniformly dispersed using magnetic stirring to form a homogeneous solution, yielding a photolithography ink material with yellow RTP emission. Similarly, we incorporated 150 mg of photopolymerizable resin into the P-AA-AM/RhB precursor solution to formulate a red RTP photolithographic ink material.

### Preparation of 3D bulk RTP materials

The 3D bulk material with RTP emission was prepared by layer-by-layer photocuring method. Specifically, the precursor solution of NDIAM-AA-AM was poured into a silicone mold, and after photopolymerization, a thin layer with yellow RTP emission was obtained. Repeating this operation yields a 3D bulk yellow RTP material. Similarly, using the precursor solution of NDIAM-AA-AM/RhB results in a 3D bulk red RTP material. Moreover, by alternately stacking and curing the two

precursors solutions layer by layer, a 3D bulk material with multicolor RTP was fabricated.

### Preparation of afterglow cotton yarn

The cotton yarn was immersed in a precursor (AA 400 μL, AM 500 mg, NDIAM 1 mg) solution. Then, the treated cotton thread was placed in 365 nm UV light for 3 min to obtain a yellow afterglow cotton thread. In order to obtain the afterglow cotton line emitted by red afterglow, 3 mg RhB solution can be added to the precursor solution for the same operation.

### Preparation of photolithography RTP patterns

The photolithography ink material is uniformly coated on the substrate, and then covered with a layer of PVC film to form a thin and uniform photolithography ink layer between the substrate and the PVC film.

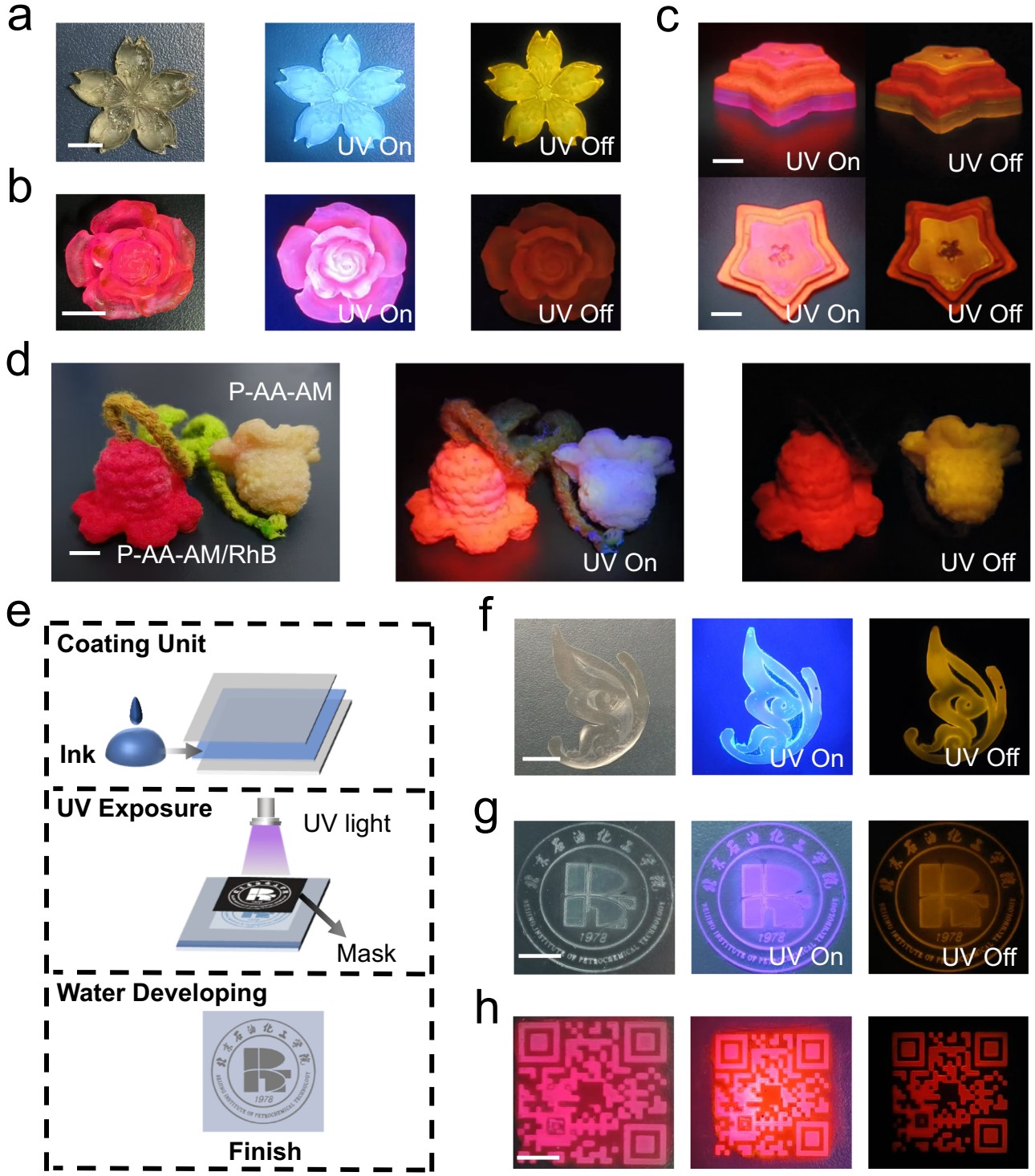

**Fig. 6 | Application of P-AA-AM and P-AA-AM/RhB. a, b** The images of 3D bulk materials made from P-AA-AM and P-AA-AM/RhB under sunlight (left), UV light (middle) and turning off UV light (right); **c** The images of 3D bulk materials made from P-AA-AM and P-AA-AM/RhB, captured as side-view (top) and top-view (bottom) under UV light (left) and turning off UV light (right); **d** Afterglow yarns after P-AA-AM and P-AA-AM/RhB treatment in sunlight, UV light and switching off the UV light; **e** Photolithography schematic of RTP material; **f** Butterfly RTP pattern on PVC film; **g** RTP pattern of school badge was fabricated on glass; **h** RTP pattern of school QR code made on glass. Scale bar: 8 mm.

Subsequently, 365 nm UV light is used to expose the designated area of the photolithography ink layer through a mask of a specific shape. After 80 s exposure, the unexposed ink was removed by developing with water, the underexposed ink was still retained after water developing. Finally, the desired pattern is formed on the substrate.

## Data availability

All relevant data are included in this article and its Supplementary Information files. All data underlying this study are available from the corresponding author Bing Fang upon request. Source data are provided with this paper.

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

## Acknowledgements

M.Y. wishes to thank the National Natural Science Foundation of China (No. 52130309 and W2412081), B.F. wishes to thank the Zhiyuan Science Foundation of Beijing Institute of Petrochemical Technology (No. 2024106), and Y.D. wishes to thank the Undergraduate Research Training Program of Beijing Institute of Petrochemical Technology (No. 2025J00211). This work was supported by the High Performance Computing Platform of BUCT.

## Author contributions

Conceptualization: B.F., M.Y., and Y.D.; Methodology: A.W. and H.W.; Investigation: A.W., H.W., K.L, X.H., M.C., W.B.; Visualization: A. W., J.W., Y.Q., Y.Z., and J.Y.; Supervision: B.F.; Writing-original draft: B. F.; Writing–review and editing: All authors.

## Competing interests

The authors declare no competing interests.
