## [Transparent Peer Review file · Nature Communications]

Proton Transfer Regulated Photocured Robust Room-Temperature Phosphorescence from Naphthalimide

Corresponding Author: Dr Bing Fang

Version 0:

Reviewer comments:

Reviewer #1

(Remarks to the Author)

Recommendation: May be suitable for Nature Communications after minor revisions.

The authors report photocured room-temperature phosphorescence materials and naphthalimide simultaneously acts as phosphor and photo-initiator. The resulting naphthalimide is tightly restricted in in-situ formed crosslinked matrix to achieve robust RTP with a long lifetime of 389.58 ms, high phosphorescence quantum yield of 17.83%, water and organic solvents resistance. Carboxyl can bind onto the lone-pair electrons of tertiary amine through PTHBs, inhibiting non-irradiation transition of S1 induced by PET; increasing SOC matrix elements values to promote ISC; affording rigid microenvironment to stabilize triplet excitons. Moreover, the afterglow colors are continuously tuned from yellow to red after loading different mass RhB via energy transfer. The as prepared materials are used as phosphorescent inks for fabricating 3D printing and photopatterning applications. This work is conceptually promising and the authors have demonstrated it through a range of experiments. I recommend the publication of this manuscript after minor revision.

Question 1. The authors declared that "NDIAM are optimized with simplified homopolymers through molecular dynamics simulation". The calculations details and methods should be added in the experiments section.

Question 2. The authors should indicate the purity of NDIAM with HPLC analysis.

Question 3. In Figure 3a, AA photo-copolymerizes with St, MA, MMA and HEA, why the afterglow photographs of P-AA-St, P-AA-MA and P-AA-MMA are inhomogeneous?

Question 4. The authors only provide the double bond conversion rate in P-AA-AM. The double bond conversion rates for other photo-polymerization systems also should be added, which can better explain the different phosphorescent properties between these polymers.

Question 5. P-AA-AM showed robust RTP with water resistance within 240 min. When removing the water, can RTP property restore to its original state?

Question 6. The authors should include more refs in introduction to show the benefits of (Controlled) radical polymerization for the synthesis of RTP polymers.

Question 7. As controlled radical polymerization could synthesize linear polymers or polymer networks with better uniformity, what will happen if controlled Radical polymerization was employed for the 3D-printing of RTP polymers?

Reviewer #2

(Remarks to the Author)

In the manuscript entitled "Proton Transfer Regulated Photocured Robust Room-Temperature Phosphorescence from Naphthalimide". Authors have designed photocured robust RTP materials with long lifetime and high phosphorescence quantum yield. The present system is also highly resistant towards water and organic solvents, as clear from long afterglow even in the presence of water. Authors also have performed extensive investigation to reveal PTHBs regulated RTP systems, such as photo-polymerize monomers without active hydrogen atom, photo-copolymerize these monomers with AA, and mixed with PVA. Moreover, yellow to red afterglow colors are continuously tuned after loading different mass RhB via TS-FRET strategy. Using these properties, the as prepared materials are used as RTP inks for fabricating 3D printing and photopatterning for anti-counterfeiting and information encryption applications. The manuscript has enough novelty and

suitable to publish in this journal.

However, the author should address the following points.

1. Authors use 1 mg photo-initiator to photo-polymerize monomers and has good RTP performance. What is the effect of different content photo-initiator on RTP property?
2. In line 177-179, authors have claimed that the “the stability of P-AA-AM is further investigated in organic solvents and RTP lifetime is not compromised when it is immersed into different organic solvents”. Please provide phosphorescence spectra after organic solvents treatment to further demonstrate the stability of RTP in P-AA-AM.
3. The basic properties of P-AA-AM need to be investigated, such as FTIR, molecular weight, XRD, T_g, T_m and mechanical property characterization.
4. In Figure 5c, the insert pictures are too small to be clearly seen. Please provide high resolution pictures.
5. Authors should mention the delay time used for delayed emission spectra in Figure 2d, 2e, and 3b.

Reviewer #3

(Remarks to the Author)

The manuscript presents a novel approach to photocurable room-temperature phosphorescence (RTP) materials using naphthalimide derivatives as both photoinitiators and phosphorescent chromophores. The RTP is convincing and can be tuned from yellow to red by varying RhB content, enabling applications in 3D printing, and information encryption. However, several mechanistic claims require stronger experimental support. Some of the major comments are listed below;

1. Similar strategies using hydrogen-bonded hosts to activate 1,8-naphthalimide RTP have been reported (e.g., Chem. Commun., 2022, 58, 3641; DOI:10.1039/d2cc00474g). The authors should cite this work and clearly explain how their system and mechanistic insights differ.
 2. The authors propose that PTHB between the tertiary amine and carboxyl groups plays a central role in regulating RTP. However, this claim requires direct experimental evidence. Please provide spectroscopic confirmation, such as FT-IR, XPS N 1s or pH-dependent absorption, to distinguish PTHB from ordinary hydrogen bonding.
 3. The manuscript claims that PTHB suppresses PET and nonradiative S₁ decay. This should be experimentally verified. Techniques such as time-resolved fluorescence or transient absorption spectroscopy could directly demonstrate suppression of nonradiative pathways and increased ISC efficiency.
 4. Can the authors further demonstrate that the acidic environment enhances triplet exciton generation and stabilization? For example, comparison of oxygen sensitivity, and low-temperature phosphorescence between acidic and non-acidic systems in addition to the phosphorescence lifetimes would clearly clarify whether the acid-amine interaction promotes triplet formation, stabilizes the triplet state, or both.
 5. The current results do not clearly distinguish the chemical effect of protonation from the physical effect of matrix rigidification. Please decouple these effects, e.g., by examining protonated NDIAM in rigid but non-polymeric matrices, or in flexible acidic polymers.
 6. The manuscript frequently uses “nonradiative transition” to describe S₁ decay pathways. Please replace this with the standard term “nonradiative decay.”
 7. Minor corrections such as, use the proper symbol for μ , not “u,” for all volume measurements (e.g., μL), carefully check spelling and grammar throughout the manuscript. Also, Figures S7 (and related spectra) would be more informative if all relevant NMR spectra are presented in a single figure for direct comparison.
- Overall, the work is promising, but mechanistic claims need stronger experimental support. I recommend major revision to address the points above. The manuscript will be significantly improved once these issues are addressed.

Version 1:

Reviewer comments:

Reviewer #1

(Remarks to the Author)

The authors have addressed all my questions. The current manuscript is recommended to be accepted.

Reviewer #2

(Remarks to the Author)

The present work can be published in Nature Communications.

Reviewer #3

(Remarks to the Author)

The reviewers' comments have been addressed satisfactorily, and the manuscript is recommended for acceptance

REVIEWER COMMENTS

Reviewer #1 (Remarks to the Author):

Recommendation: May be suitable for Nature Communications after minor revisions. The authors report photocured room-temperature phosphorescence materials and naphthalimide simultaneously acts as phosphor and photo-initiator. The resulting naphthalimide is tightly restricted in in-situ formed crosslinked matrix to achieve robust RTP with a long lifetime of 389.58 ms, high phosphorescence quantum yield of 17.83%, water and organic solvents resistance. Carboxyl can bind onto the lone-pair electrons of tertiary amine through PTHBs, inhibiting non-irradiation transition of S_1 induced by PET; increasing SOC matrix elements values to promote ISC; affording rigid microenvironment to stabilize triplet excitons. Moreover, the afterglow colors are continuously tuned from yellow to red after loading different mass RhB via energy transfer. The as prepared materials are used as phosphorescent inks for fabricating 3D printing and photopatterning applications. This work is conceptually promising and the authors have demonstrated it through a range of experiments. I recommend the publication of this manuscript after minor revision.

Question 1. The authors declared that “NDIAM are optimized with simplified homopolymers through molecular dynamics simulation”. The calculations details and methods should be added in the experiments section.

Response: Thank you for pointing out this. All molecular conformations optimized and vibrational frequency analyses were performed via the Gaussian 09 program at the B3LYP/6-31G(d,p) level of theory. Excited-state calculations were carried out using the ORCA 4.2.0 program at the PBE0/def2-SV(P) level. Analysis of the electrostatic potential surfaces (ESPs), independent gradient model based on Hirshfeld partition (IGMH) analysis, and natural transition orbital (NTO) analysis for the singlet (S_1) and triplet (T_1) excited states were completed with the Multiwfn_3.8_dev program, and the corresponding plots were generated using the VMD 1.9.3 program. The calculations details and methods have been added in the revised manuscript (Page 15, Line 14-16; Supplementary Information, Page S5, Line 4-13).

Question 2. The authors should indicate the purity of NDIAM with HPLC analysis.

Response: We have performed HPLC to ensure the high purity of NDIAM compounds and have been added in the revised manuscript. (Supplementary Fig. 4; Supplementary Table 1).

Question 3. In Figure 3a, AA photo-copolymerizes with St, MA, MMA and HEA, why the afterglow photographs of P-AA-St, P-AA-MA and P-AA-MMA are inhomogeneous?

Response: Probably, the energy of UV light is not uniform, thereby leading to afterglow photographs of P-AA-St, P-AA-MA and P-AA-MMA are inhomogeneous. We re-test the photo-polymerization of AA with St, MMA, and MA and homogeneous afterglow photographs have been updated in the revised manuscript (Page 14, Figure 3a; Supplementary Fig. S24-S27).

Question 4. The authors only provide the double bond conversion rate in P-AA-AM. The double bond conversion rates for other photo-polymerization systems also should be added, which can better explain the different phosphorescent properties between these polymers.

Response: Thank you very much for your valuable comment. The double bond conversion rate reaches ~ 22.22%, 66.03%, 84.44%, 69.07%, and 61.78% in P-St, P-MA, P-MMA, P-HEA, and P-AA, respectively, after 365 nm UV light irradiation for 300 s (Power intensity: 300 mW/cm²). The conversion rate of P-AA is not the highest in these polymers, however, P-AA shows the best RTP performance, which are due to proton transfer hydrogen-bonds and intermolecular H-bonds immobilize NDIAM and stabilize triplet excitons. These contents have been added in the revised manuscript (Page 10, Line 11-14; Supplementary Fig. 21 and Supplementary Fig. 28).

Question 5. P-AA-AM showed robust RTP with water resistance within 240 min. When removing the water, can RTP property restore to its original state?

Response: The RTP lifetime and intensity of P-AA-AM can restore to its original state after removing the water (Supplementary Fig. 17).

Question 6. The authors should include more refs in introduction to show the benefits of (Controlled) radical polymerization for the synthesis of RTP polymers.

Response: Thank you for your comments. The related reference about controlled radical polymerization for the synthesis of RTP polymers have been added in the revised manuscript (Page 3, Line 2; Page 29, Line 23-29, Reference: 31-33).

Question 7. As controlled radical polymerization could synthesize linear polymers or polymer networks with better uniformity, what will happen if controlled Radical polymerization was employed for the 3D-printing of RTP polymers?

Response: Thank you very much for your valuable comment. Controlled radical polymerization could synthesize linear polymers or polymer networks with better uniformity, however, it has very high requirements for the catalyst and is sensitive to the environment (H_2O and O_2), which can't directly apply in 3D-printing in current stage. If controlled radical polymerization is employed for the 3D-printing RTP polymers, the process parameters of 3D printing and RTP property is pretty stable and good repeatability.

Reviewer #2 (Remarks to the Author):

In the manuscript entitled “Proton Transfer Regulated Photocured Robust Room-Temperature Phosphorescence from Naphthalimide”. Authors have designed photocured robust RTP materials with long lifetime and high phosphorescence quantum yield. The present system is also highly resistant towards water and organic solvents, as clear from long afterglow even in the presence of water. Authors also have performed extensive investigation to reveal PTHBs regulated RTP systems, such as photo-polymerize monomers without active hydrogen atom, photo-copolymerize these monomers with AA, and mixed with PVA. Moreover, yellow to red afterglow colors are continuously tuned after loading different mass RhB via TS-FRET strategy. Using these properties, the as prepared materials are used as RTP inks for fabricating 3D printing and photopatterning for anti-counterfeiting and information encryption applications. The manuscript has enough novelty and suitable to publish in this journal.

However, the author should address the following points.

Question 1. Authors use 1 mg photo-initiator to photo-polymerize monomers and has good RTP performance. What is the effect of different content photo-initiator on RTP property?

Response: The influence of the doping mass of NDIAM on phosphorescent intensity and lifetime was also investigated. 0.01, 0.1, 1, 5, and 10 mg NDIAM were used to photo-polymerize AA and AM. Phosphorescence intensity and lifetime reach a maximum at doping 1 mg NDIAM. It could be concluded that excess photo-initiator resulted in energy dissipation and aggregation-caused quenching. The related contents have been added in the revised manuscript (Page 7, Line 16-21; Supplementary Fig. 15).

Question 2. In line 177-179, authors have claimed that the “the stability of P-AA-AM is further investigated in organic solvents and RTP lifetime is not compromised when it is immersed into different organic solvents”. Please provide phosphorescence spectra after organic solvents treatment to further demonstrate the stability of RTP in P-AA-AM.

Response: Phosphorescence intensity after organic solvents treatment is not compromised when compares with the initial state (Supplementary Fig. 19).

Question 3. The basic properties of P-AA-AM need to be investigated, such as FTIR, molecular weight, XRD, T_g , T_m and mechanical property characterization.

Response: Thank you very much for your valuable comment. The FTIR, XRD, T_g , and mechanical property characterization have been tested. Many solvents (water, THF, DMF and DMSO) can't dissolved P-AA-AM even heated to 180 °C. It might be due to the formation of 3D polymer network during the photo-polymerization process. So, molecular weight can't be investigated at the current stage. XRD pattern indicates that P-AA-AM adopts amorphous form and with no definite T_m . The related contents have been added in the revised manuscript (Page 6, Line 29; Page 7, Line 1-3; Supplementary Figs. 9-10).

Question 4. In Figure 5c, the insert pictures are too small to be clearly seen. Please provide high resolution pictures.

Response: Thank you for pointing out this. High resolution pictures have been updated in the revised manuscript (Page 21, Figure 5c).

Question 5. Authors should mention the delay time used for delayed emission spectra in Figure 2d, 2e, and 3b.

Response: The delay time is 1 ms for delayed emission spectra in Figure 2d, 2e, and 3b (Page 9, Line 6-8; Page 14, Line 4).

Reviewer #3 (Remarks to the Author):

The manuscript presents a novel approach to photocurable room-temperature phosphorescence (RTP) materials using naphthalimide derivatives as both photoinitiators and phosphorescent chromophores. The RTP is convincing and can be tuned from yellow to red by varying RhB content, enabling applications in 3D printing, and information encryption. However, several mechanistic claims require stronger experimental support. Some of the major comments are listed below.

Question 1. Similar strategies using hydrogen-bonded hosts to activate 1,8-naphthalimide RTP have been reported (e.g., *Chem. Commun.*, 2022, 58, 3641; DOI:10.1039/d2cc00474g). The authors should cite this work and clearly explain how their system and mechanistic insights differ.

Response: Thank you very much for your valuable comment. This work reports the room-temperature phosphorescence of 1,8-naphthalimide was activated by doping it into aromatic dicarboxylic acids (*Chem. Commun.*, 2022, 58, 3641; DOI:10.1039/d2cc00474g). Abundant intermolecular hydrogen-bond network can be constructed through interactions between carboxyl groups of aromatic dicarboxylic acids and imidyl groups of 1,8-naphthalimide, which can suppress the non-radiative decay process to activate room-temperature phosphorescence. 1,8-naphthalimide doped into other hosts, such as PVA, the RTP could also be activated (*Macromol. Chem. Phys.* 2025, 226, 2400498). However, in our system, NDIAM exhibits no afterglow when

doped into PVA matrix, or photo-polymerize St, MA, MMA and HEA. Long lifetime RTP cannot be realized by intermolecular hydrogen bonds alone. Bright afterglows are obtained after adding acidic compounds into above systems. Together with time-resolved fluorescence, pH-dependent absorption, and theory calculations, proton-transfer hydrogen bonds (PTHBs) between acid-amine are formed and play as following roles: (1) inhibiting non-irradiation transition of S_1 state induced by PET; (2) increasing SOC values to promote ISC; (3) providing a rigid microenvironment that effectively immobilizing NDIAM motion to stabilize triplet excitons for phosphorescence emission. We have cited this work in the revised manuscript (Page 29, Line 11-13, Reference 27).

Question 2. The authors propose that PTHB between the tertiary amine and carboxyl groups plays a central role in regulating RTP. However, this claim requires direct experimental evidence. Please provide spectroscopic confirmation, such as FT-IR, XPS N 1s or pH-dependent absorption, to distinguish PTHB from ordinary hydrogen bonding.

Response: Thank you very much for your valuable comment. NDIAM solution (2×10^{-4} M, MeOH) was mixed with different volumes of AA solution (2×10^{-3} M, MeOH), making NDIAM and AA at different molar weight. The absorption peak at ~ 235 nm exhibits gradually reduced trend upon adding AA (Supplementary Fig. 36). AA, NDIAM-AA (150 mg NDIAM and 600 μ L AA), and triethylamine (TEA)-AA (200 μ L TEA and 600 μ L AA) of FTIR were also tested. Upon adding NDIAM and triethylamine (TEA) into AA solution, the vibration peaks of the hydroxyl at 2992 cm^{-1} and the carbonyl at 1694 cm^{-1} in carboxyl weaken; while the vibration peaks of the asymmetric and symmetric the carboxylate at 1541 cm^{-1} and 1355 cm^{-1} appear (Supplementary Fig. 37). The hydrogen atoms in methyl group of NDIAM, exhibits significant chemical shift upon adding AA (Supplementary Fig. 38). Together with pH-dependent absorption, FT-IR, and ^1H NMR, these changes upon NDIAM meeting with AA, show the formations of proton transfer hydrogen-bonds between the tertiary amine and carboxyl groups. The contents have been added in the revised manuscript (Page 15, Line 2-13;

Supplementary Figs. 36-38).

Question 3. The manuscript claims that PTHB suppresses PET and nonradiative S_1 decay. This should be experimentally verified. Techniques such as time-resolved fluorescence or transient absorption spectroscopy could directly demonstrate suppression of nonradiative pathways and increased ISC efficiency.

Response: Thank you very much for your valuable comment. Time-resolved fluorescence tests revealed the differences in the luminescence dynamics of P-AA-St and P-St. Both P-AA-St and P-St exhibited double-exponential decay characteristics, but their excited-state lifetime distributions were significantly different. For P-AA-St, the fluorescence lifetime (τ_1 , 1.77 ns, accounting for 45.29%) was significantly longer than that of P-St (τ_1 , 0.87 ns, accounting for 76.01%), and its long lifetime component (τ_2) contribution was as high as 54.71%, much higher than 23.99% of P-St. These results indicate that the excited-state lifetime in P-AA-St was significantly prolonged, and nonradiative S_1 decay is effectively suppressed, thereby enhancing the fluorescence emission. The contents have been added in the revised manuscript (Figure 3h-3j; Page 11, Line 22-27; Supplementary Fig. 29).

Question 4. Can the authors further demonstrate that the acidic environment enhances triplet exciton generation and stabilization? For example, comparison of oxygen sensitivity, and low-temperature phosphorescence between acidic and non-acidic systems in addition to the phosphorescence lifetimes would clearly clarify whether the acid-amine interaction promotes triplet formation, stabilizes the triplet state, or both.

Response: Thank you very much for your valuable comment. Under vacuum condition, in non-acidic systems (NDIAM/PVA), phosphorescence intensity and lifetime are too low to detect; in acidic system (NDIAM-AA/PVA), strong phosphorescence intensity and 210.18 ms lifetime are obtained, which are longer than that of in air (Supplementary Fig. 32). Consequently, it was not oxygen that quenched the phosphorescence of NDIAM/PVA. Under low-temperature condition (77K), NDIAM/AA/PVA shows higher phosphorescence intensity and longer lifetime than that of NDIAM/PVA

(Supplementary Figs. 33, 34). These results indicate that proton transfer hydrogen-bonds between acid-amine promoting triplet formation and stabilizing triplet state for phosphorescence emission. Related contents have been added in the revised manuscript (Page 12, Line 21-29; Page 13, Line 1-3).

Question 5. The current results do not clearly distinguish the chemical effect of protonation from the physical effect of matrix rigidification. Please decouple these effects, e.g., by examining protonated NDIAM in rigid but non-polymeric matrices, or in flexible acidic polymers.

Response: Thank you very much for your valuable comment. Protonated NDIAM was prepared by adding hydrogen chloride solution into NDIAM DCM solution, and precipitate (NDIAM-H) was obtained. We doped NDIAM-H into γ -cyclodextrin (γ -CD) and PAA. NDIAM-H/PAA shows higher phosphorescence intensity and longer lifetime than that of NDIAM-H/ γ -CD, indicating that physical effect of matrix rigidification also play an important role in regulating RTP performance (Page 13, Line 3-10; Supplementary Fig. 35).

Question 6. The manuscript frequently uses “nonradiative transition” to describe S_1 decay pathways. Please replace this with the standard term “nonradiative decay.”

Response: Thank you for pointing out this. We have replaced "nonradiative transition" with "non-radiative decay" in the revised manuscript (Page 2, Line 2; Page 3, Line 30; Page 4, Line 1, 16; Page 7, Line 6, 9, and 16; Page 10, Line 28; Page 16, Line 6, 11 and 15; Page 17, Line 9 and 13).

Question 7. Minor corrections such as, use the proper symbol for μ , not “u,” for all volume measurements (e.g., μL), carefully check spelling and grammar throughout the manuscript. Also, Figures S7 (and related spectra) would be more informative if all relevant NMR spectra are presented in a single figure for direct comparison.

Overall, the work is promising, but mechanistic claims need stronger experimental support. I recommend major revision to address the points above. The manuscript will

be significantly improved once these issues are addressed.

Response: Thank you for pointing out this. We have modified these numerous grammar errors and carefully polishing the language in the revised manuscript (Page 6, Line 12; Page 24, Line 22; Page 25, Line 3). Relevant NMR spectra are presented in a single figure for direct comparison (Supplementary Fig. 8).

REVIEWERS' COMMENTS

Reviewer #1 (Remarks to the Author):

The authors have addressed all my questions. The current manuscript is recommended to be accepted.

Reviewer #2 (Remarks to the Author):

The present work can be published in Nature Communications.

Reviewer #3 (Remarks to the Author):

The reviewers' comments have been addressed satisfactorily, and the manuscript is recommended for acceptance.

Response to reviewers:

Dear Reviewers,

Thank you very much for your positive review and for agreeing to accept our manuscript. We are truly grateful for your time and valuable feedback, which has greatly contributed to the quality of our work.

We confirm that we have carefully reviewed the final version of the manuscript. All the revisions and improvements suggested during the review process have been implemented, and we believe the manuscript now meets the journal's standards.

Once again, thank you for your support and for helping us bring this work to publication. We look forward to the next steps in the publication process.

Sincerely,

Bing Fang, Meizhen Yin, and Yuhua Dai